

# Advances in the study of nodavirus

Chean Yeah Yong[1], Swee Keong Yeap[2], Abdul Rahman Omar[1] and Wen Siang Tan[1,3]

[1] Institute of Bioscience, Universiti Putra Malaysia, Serdang, Selangor, Malaysia
[2] Xiamen University Malaysia, Sepang, Selangor, Malaysia
[3] Department of Microbiology, Faculty of Biotechnology and Biomolecular Sciences, Universiti Putra Malaysia, Serdang, Selangor, Malaysia

Corresponding author
Wen Siang Tan, wstan@upm.edu.my

## ABSTRACT

Nodaviruses are small bipartite RNA viruses which belong to the family of *Nodaviridae*. They are categorized into alpha-nodavirus, which infects insects, and beta-nodavirus, which infects fishes. Another distinct group of nodavirus infects shrimps and prawns, which has been proposed to be categorized as gamma-nodavirus. Our current review focuses mainly on recent studies performed on nodaviruses. Nodavirus can be transmitted vertically and horizontally. Recent outbreaks have been reported in China, Indonesia, Singapore and India, affecting the aquaculture industry. It also decreased mullet stock in the Caspian Sea. Histopathology and transmission electron microscopy (TEM) are used to examine the presence of nodaviruses in infected fishes and prawns. For classification, virus isolation followed by nucleotide sequencing are required. In contrast to partial sequence identification, profiling the whole transcriptome using next generation sequencing (NGS) offers a more comprehensive comparison and characterization of the virus. For rapid diagnosis of nodavirus, assays targeting the viral RNA based on reverse-transcription PCR (RT-PCR) such as microfluidic chips, reverse-transcription loop-mediated isothermal amplification (RT-LAMP) and RT-LAMP coupled with lateral flow dipstick (RT-LAMP-LFD) have been developed. Besides viral RNA detections, diagnosis based on immunological assays such as enzyme-linked immunosorbent assay (ELISA), immunodot and Western blotting have also been reported. In addition, immune responses of fish and prawn are also discussed. Overall, in fish, innate immunity, cellular type I interferon immunity and humoral immunity co-operatively prevent nodavirus infections, whereas prawns and shrimps adopt different immune mechanisms against nodavirus infections, through upregulation of superoxide anion, prophenoloxidase, superoxide dismutase (SOD), crustin, peroxinectin, anti-lipopolysaccharides and heat shock proteins (HSP). Potential vaccines for fishes and prawns based on inactivated viruses, recombinant proteins or DNA, either delivered through injection, oral feeding or immersion, are also discussed in detail. Lastly, a comprehensive review on nodavirus virus-like particles (VLPs) is presented. In recent years, studies on prawn nodavirus are mainly focused on *Macrobrachium rosenbergii* nodavirus (*Mr*NV). Recombinant *Mr*NV VLPs have been produced in prokaryotic and eukaryotic expression systems. Their roles as a nucleic acid delivery vehicle, a platform for vaccine development, a molecular tool for mechanism study and in solving the structures of *Mr*NV are intensively discussed.

## INTRODUCTION

The current review discusses recent studies related to nodaviruses. Recent reported outbreaks of nodaviruses, diagnostic assays, host immunological responses, vaccines, and virus-like particles (VLPs) are emphasized. To the best of our knowledge, there are only eight review articles related to nodavirus which had been published within the past five years: immunological-based detection of shrimp viruses (*Chaivisuthangkura, Longyant & Sithigorngul, 2014*); recombinant nodavirus-like particles as delivery system (*Jariyapong, 2015*); the life cycle of beta-nodaviruses (*Low et al., in press*); viral encephalopathy and retinopathy in aquaculture (*Doan et al., 2017*); interaction between beta-nodavirus and its host for development of prophylactic measures for viral encephalopathy and retinopathy (*Costa & Thompson, 2016*); reactive oxygen species-mediated cell death (*Reshi, Su & Hong, 2014*); mitochondrial disruption and necrotic cell death (*Hong, 2013*); and immunity to beta-nodavirus infections of marine fish (*Chen, Wang & Chen, 2014*). Another two review articles published within the past 10 years are about the biology and biomedical applications of Flock House virus (*Venter & Schneemann, 2008*), and white-tail-disease (WTD) in *Macrobrachium rosenbergii* (*Bonami & Sri Widada, 2011*). However, none of these articles review the recent advances in the study of nodaviruses as presented in the current review.

## SURVEY METHODOLOGY

"PubMed" and "Scopus" were used to search for journal articles published within the last five years using the keyword "nodavirus". These articles were screened and used as references for the current review. Additional information was obtained through the "Google" search engine with more specific keywords for older publications.

### Nodavirus

Nodavirus belongs to the family of *Nodaviridae*. Generally, nodaviruses are classified into alpha-nodavirus and beta-nodavirus based on their hosts. Alpha-nodaviruses such as Nodamura virus (NoV), Flock House virus (FHV), black bettle virus (BBV), Pariacoto virus (PaV), and a recently discovered mosinovirus (MoNV) (*Schuster et al., 2014*) infect insects, whereas beta-nodaviruses such as striped jack nervous necrosis virus (SJNNV), barfin flounder nervous necrosis virus (BFNNV), redspotted grouper nervous necrosis virus (RGNNV), and tiger puffer nervous necrosis virus (TPNNV) infect fishes. Another type of nodavirus infects prawn, and is distinctive from alpha- and beta-nodaviruses (*Naveen Kumar et al., 2013*). This prawn nodavirus includes *Macrobrachium rosenbergii* nodavirus (*Mr*NV) and *Penaeus vannamei* nodavirus (*Pv*NV). *Naveen Kumar et al. (2013)* proposed that the *Mr*NV and *Pv*NV should be categorized into gamma-nodaviruses, based on their distinct genomic sequences compared with that of both the alpha- and beta-nodaviruses. More recent studies have identified another two prawn nodaviruses, namely the covert mortality nodavirus (CMNV) (*Zhang et al., 2014*; *Zhang et al., in press*) and *Farfantepenaeus duorarum* nodavirus (*Fd*NV) (*Ng et al., 2013*), infecting *Litopenaeus vannamei* and *F. duorarum*, respectively. Although nodaviruses are usually named after

their native hosts, nodaviruses are often capable of infecting multiple species. RGNNV has been reported to infect Asian seabass, *Lates calcarifer* (*Banerjee et al., 2014*); Nile tilapia, *Oreochromis niloticus* (*Keawcharoen et al., 2015*); and *Amphiprion sebae* Bleeker, a marine clownfish (*Binesh et al., 2013*), whereas *Mr*NV has also been reported to infect *Penaeus indicus*, *Penaeus monodon*, and *P. vannamei* (*Ravi et al., 2009*; *Senapin et al., 2012*).

## Fish nodavirus

The fish nodavirus, also known as Nervous Necrosis Virus (NNV), infects fishes and causes viral encephalopathy and retinopathy (VER). The first outbreak occurred in 1985 (*Costa & Thompson, 2016*). The infection was first described by *Yoshikoshi & Inoue (1990)* in the Japanese parrot fish *Oplegathus fasciatus*. Fishes infected by nodavirus suffer neurological disorders, which are characterized by intensive vacuolization of retina and central nervous systems, culminating in abnormal swimming pattern and darkening of fish color (*Munday & Nakai, 1997*). In fish, Nodavirus can be detected in many organs but central nervous system including the brain, spinal cord and retina are the main targets (*Ghiasi et al., 2016*). The fish nodavirus seriously affects aquaculture industry worldwide, resulting in great economic losses. Infection by this virus is often associated with high mortality rate, up to 100% in fish larvae and juveniles (*Skliris et al., 2001*). To date, nodavirus is known to affect over 120 fish species, particularly groupers and seabass such as the Asian seabass *Lates calcarifer* and European seabass *Dicentrarchus labrax* (*Breuil et al., 1991*; *Costa & Thompson, 2016*; *Frerichs, Rodger & Peric, 1996*; *Munday, Kwang & Moody, 2002*; *Parameswaran et al., 2008*). Although nodavirus mostly affects marine fishes, nodavirus infections in freshwater fishes such as European eels (*Anguilla anguilla* L.), yellow-wax pompano (*Trachinotus falcatus*), firespot snapper (*Lutaanus erythropterus* B.), cobia (*Rachycentron canadum*) and Chinese catfish (*Parasilurus asotus*) have been reported in Taiwan (*Chi, Shieh & Lin, 2003*). In addition, outbreaks in hybrid striped bass × white bass (*Morone saxalitis × Morone chrysops*) and largemouth bass (*Micropterus salmoides*) have also been reported in Italy (*Bovo et al., 2011*). Apart from horizontal transmission, fish nodavirus can be transmitted vertically through infections at the gonads, passing the virus to their progenies (*Breuil et al., 2002*; *Kocan, Hershberger & Elder, 2001*; *Valero et al., 2015a*).

## Prawn nodavirus

Like fish nodavirus, prawn nodavirus has significant economic impact on the prawn aquaculture industry. Prawn nodavirus can be isolated from cephalothoraxes and whitish abdominal muscle (*Zhang et al., 2014*) of infected prawns. The most studied prawn nodavirus is the *Mr*NV. It is a non-zoonotic nodavirus which infects *M. rosenbergii*, commonly known as the giant river prawn. *Mr*NV was first isolated and reported in 1999 (*Arcier et al., 1999*) from *M. rosenbergii*. Infection by *Mr*NV causes white tail disease (WTD) or white muscledisease (WMD), where infected cells undergo necrosis and turn whitish. The rate of mortality is extremely high (up to 100%) in larvae and post-larvae of *M. rosenbergii* (*Qian et al., 2003*; *Ravi et al., 2009*), causing great economic losses to *M. rosenbergii* hatchery and nursery farm industries. Despite the high mortality rate in larvae and post-larvae prawns, *Mr*NV does not cause death in adult prawns.

However, the adult prawns still serve as the virus carriers, transmitting the virus vertically to their offsprings (*Sudhakaran et al., 2007*), and horizontally to other prawns during cannibalization (*Sahul Hameed et al., 2004*). Another prawn virus, *Pv*NV was first isolated in 2005 from a *P. vannamei* farm in Belize (*Tang et al., 2007*; *Tang et al., 2011*). Being a prawn nodavirus, *Pv*NV shares 83% similarities with *Mr*NV in its viral genome (*Tang et al., 2011*). It causes muscle necrosis, resulting in white, opaque lesions in the tail, similar to the symptoms of *Mr*NV infection. However, the virulence of *Pv*NV is not as high as *Mr*NV, in which the former normally resulted in approximately 50% production loss in an infected farm (*Tang et al., 2007*). Apart from its native host, *Pv*NV has also been demonstrated to be able to infect *Penaeus monodon* in an experimental infection (*Tang et al., 2007*).

## Insect nodavirus

Unlike fish and prawn nodaviruses, insect nodaviruses do not have a direct impact on the global economy. Despite that, insect nodaviruses, especially the FHV and BBV, have served as excellent models to study the mechanisms of other positive-strand RNA viruses, such as those of *Caliciviridae*, *Flaviviridae*, *Picornaviridae*, and *Togaviridae*, due to their small genome size and high level of replication in compatible hosts (*Ball & Johnson, 1998*). FHV was originated from grass grub, *Costelytra zealandica* (*Dearing et al., 1980*). FHV has been demonstrated to be able to infect a wide variety of hosts, including insects, yeasts, plants, and mammalian cells. Apart from its original host *C. zealandica*, FHV also infects the common fruit fly, *Drosophila melanogaster*. Therefore, cell-lines derived from *D. melanogaster* such as Schneider Line 1 (DL1) have been established for the propagation of insect nodaviruses (*Dearing et al., 1980*; *Miller, Schwartz & Ahlquist, 2001*). When the yeast *Saccharomyces cerevisiae* was transfected with FHV, the viral genomic RNA induced the production of infectious virion capable of infecting *Drosophila* cells (*Price, Rueckert & Ahlquist, 1996*). In addition, FHV was also reported to infect the whole plants of barley, cowpea, chenopodium and tobacco (*Selling, Allison & Kaesberg, 1990*), as well as mammalian cells such as the baby hamster kidney cell (BHK21) (*Ball, Amann & Garrett, 1992*). Due to its wide host range, FHV has been an excellent model to study the mechanisms of other economically important RNA viruses. Another well-studied insect virus, BBV, was isolated from *Heteronychus arator*. BBV propagates well in *Drosophila* line 1 cells, but not in BHK21, mouse L-cell, mosquito cells (*Aedes albopictus* and *A. aegypti*), cabbage looper (*Trichoplasia ni*), fall armyworm (*Spodoptera frugiperda*) and line GM1 of *D. melanogaster* (*Friesen et al., 1980*). *Selling & Rueckert (1984)* established a plaque assay for nodaviruses using *Drosophila* cell-adapted BBV, which greatly facilitates the isolation and reassortant of nodaviruses (*Kopek et al., 2010*; *Settles & Friesen, 2008*). BBV's structure has been studied intensively using electron microscopy and crystallization followed by small-angle x-ray scattering (*Hosur et al., 1984*). As in other nodaviruses, BBV appeared to form icosahedral structure with a triangulation number of $T = 3$. Furthermore, the RNA3 of nodavirus was identified to be a subgenomic mRNA of the viral RNA1 by studying BBV, and it can be isolated from cells infected by BBV (*Friesen & Rueckert, 1982*; *Guarino et al., 1984*). Another insect nodavirus, Boolarra virus was isolated from the ghost moth *Oncopera intricoides* (*Reinganum, Bashiruddin & Cross, 1985*). The viral morphogenesis was shown

to be restricted to the cytoplasm of cultured *Drosophila* cell lines (*Bashiruddin & Cross, 1987*). A more recent Wuhan nodavirus was isolated from *Pieris rapae* larvae (*Liu et al., 2006a*). A study of its subgenomic RNA3 has provided an insight into the RNAi inhibitory property of the nodavirus B2 protein (*Cai et al., 2010*).

## General features of nodavirus

In general, nodaviruses are non-enveloped zoonotic viruses with icosahedral structures. Their genomes comprise of two linear, positive-sense, single-stranded RNA. RNA 1 is approximately 3.1–3.2 kilobases (kb) in length, whereas RNA2 is approximately 1.2–1.4 kb. Both of which lack a poly-A tail at their 3′ ends (*Comps, Pepin & Bonami, 1994*; *Mori et al., 1992*). RNA 1 encodes for the RNA-dependent RNA polymerase (RdRP), which functions in replicating the viral RNA genome without involving an intermediate DNA. RNA 3, a subgenomic transcript of RNA 1, it encodes for a non-structural B2-like protein (*Cai et al., 2010*; *Hayakijkosol & Owens, 2012*; *Lingel et al., 2005*). B2 functions as a suppressor for the post-transcriptional gene silencing of host defense mechanisms through non-specific binding to double-stranded RNA generated during the virus replication (*Fenner et al., 2006*). RNA 2 encodes for the viral capsid protein, which forms the core of nodavirus. The nodavirus capsid protein assembles into virus particles with icosahedral structures, approximately 30 nm in diameter, with a triangulation number of 3 ($T = 3$) containing 180 capsid subunits. The virus particles package only the RNA 1 and RNA 2, forming simple but infectious virions.

## Transmission of nodavirus

It has been confirmed that vertical transmission is the main mechanism of nodavirus spreading (*Murwantoko et al., 2016*; *Zhang et al., in press*). This vertical transmission in the aquaculture industry can be overcome by good biosecurity practices in hatchery-reared larvae and juveniles of some fish species. Besides vertical transmission, nodavirus may also infect the cultured fish even at the grow-out stages through horizontal transmission. Although nodaviruses detected in aquaculture farms are often with relatively low sequence variations, PCR based molecular analyses (including RT-PCR and nested PCR) have revealed different betanodaviruses with high numbers of sequence variations in wild fishes and even seawater samples. This implies that, nodavirus with different virulence may be shed by the less susceptible wild fish in water and consequently virulent forms of nodavirus in the seawater would infect the susceptible cultured fish (*Nishi et al., 2016*).

## Recent incidence of nodavirus

Nodavirus infection has a great negative impact on the aquaculture industry. To date, more than 40 marine and freshwater fish species have been identified susceptible to nodavirus (particularly betanodavirus) infection (*Nishi et al., 2016*). It has been detected in freshwater prawn hatcheries in Indonesia (*Murwantoko et al., 2016*) and marine shrimp farms located in Fujian, Shandong and Hebei Provinces in China (*Zhang et al., 2014*). In addition, nodavirus caused mass mortality in cage-reared freshwater guppy *Poicelia reticulate* in Singapore (*Hegde et al., 2003*), larval rearing facility of marine clownfish, *Amphiprion sebae* in India (*Binesh et al., 2013*) and Asian seabass in India (*Banerjee et al., 2014*). Apart from

affecting aquaculture industry, nodaviruses detected in wild golden grey mullet *Liza aurata* and sharpnose mullets *Liza saliens* were correlated to the dramatically decrease of mullets stock in the Caspian Sea (*Zorriehzahra et al., 2014*; *Ghiasi et al., 2016*).

## Detection of nodavirus
### General identification
Histopathology and Transmission Electron Microscopy (TEM) examinations were used to observe the presence of nodaviruses in fishes (*Ghiasi et al., 2016*) and shrimps (*Zhang et al., 2014*). In terms of histopathological analysis of nodavirus infected shrimps, necrotic epithelium and inclusions in the hepatopancreatic tubular epithelium are commonly observed in a nodavirus infected shrimp. In addition, viral inclusion and viral particles are commonly observed in the hepatopancreas using TEM (*Zhang et al., 2014*). Moreover, severe anemia associated with increase of neutrophil populations, decrease of lymphocyte populations, raise of liver enzyme profile and decline of total protein, albumin and total immunoglobulin levels were also observed in fishes infected with nodaviruses (*Ghiasi et al., 2016*).

### Molecular identification
For the phylogenetic analysis, conventional and real-time reverse transcription PCR (RT-PCR) that amplify the RNA-dependent RNA polymerase (RdRp) of RNA 1 (*Murwantoko et al., 2016*) or the T4 region of RNA2 of nodavirus (*Nishizawa et al., 1994*; *Hegde et al., 2003*; *Banerjee et al., 2014*; *Overgård et al., 2012*), random shotgun metagenomic sequencing (*Ng et al., 2013*) and Illumina whole transcriptome metagenomic sequencing were able to detect the presence of nodavirus in infected organisms (*Greninger & DeRisi, 2015*) or even from seawater (*Nishi et al., 2016*). For example, tombunodavirus that shares nucleotide sequence similarity with that of nodavirus and tombuvirus family members was identified in the weekly metagenomic sequencing of organisms in San Francisco wastewater (*Greninger & DeRisi, 2015*). Nevertheless, whether this phenomenon was due to co-infection of nodavirus and tombuvirus or the real existence of tombunodavirus needs further validation. In another study by *Conceição Neto et al. (2015)*, a putative novel member of nodavirus was detected in the fecal samples of otter (*Lutra lutra*) in Portugal based on the identification of RdRp in the metagenomic analysis. However, this nodavirus identified in the gut of the otter may have originated from a fish diet, which casts doubt on the report of a new host for nodavirus (*Conceição Neto et al., 2015*).

To improve molecular identification, virus isolation coupled with either the Sanger sequencing or next generation sequencing (NGS) allow specific characterization of a particular strain of nodavirus (*Zhang et al., 2014*). Based on the International Committee on Taxonomy of Viruses (ICTV), isolated nodaviruses can be classified according to the genetic diversity of the RNA2 segment by the simple and cost-effective Sanger sequencing method (*Conceição Neto et al., 2015*). Pairwise identity of the RNA2 with less than 80% at the nucleotide level and less than 87% at the amino acid level is classified as a novel species (*Schuster et al., 2014*). Compared to the partial sequence identity determined by the Sanger sequencing method, profiling the whole transcriptome of a nodavirus offers a more comprehensive comparison and characterization of the virus classification. For example,

CMNV, an alphanodavirus that shares only 31-54% nucleotide sequence similarity with other nodaviruses in GenBank, was successfully characterized by sequencing the cDNA library using the Roche 454 sequencer (*Zhang et al., 2014*). In addition, fluorescence *in situ* hybridization (FISH) and nested RT-PCR assays that detect a specific nodavirus can be designed (*Zhang et al., 2014*). Another study by *Schuster et al. (2014)* reported that the identification of Mosinovirus (MoNV), a novel member of the family *Nodaviridae*, belongs neither to alpha- nor beta-nodaviruses. Without the isolation of the virus, recombination that was detected by the whole transcriptome 454 pyrosequencing in MoNV would not be accepted (*Schuster et al., 2014*). Although nodavirus can be detected in water samples, quantitative isolation of the nodavirus remains challenging with the current available protocols (*Nishi et al., 2016*).

## Diagnosis of nodaviruses

Most of the diagnostic assays for nodaviruses are based on detection of the viral RNA through RT-PCR. In recent years, efforts have been focused to establish diagnostic assays which require minimal laboratory setup. A rapid and sensitive automated microfluidic chip system for the detection of piscine nodavirus in groupers has been developed (*Kuo et al., 2012*). The microfluidic chip contains an RT-PCR module capable of processing extracted RNA samples, and a capillary electrophoresis module. This microchip has been field-tested in an epidemiological investigation of NNV in Taiwan (*Kuo et al., 2012*).

Reverse-transcription loop-mediated isothermal amplification (RT-LAMP) is another potential point-of-care diagnostic assay, as a laboratory setup such as thermocycler and electrophoresis equipment can be omitted. *Suebsing, Prombun & Kiatpathomchai (2013)* developed an RT-LAMP with colorimetric gold nanoparticle probe assay for the detection of PvNV in *P. vannamei* and *P. monodon*. This assay is 10x more sensitive than the nested RT-PCR established by *Tang et al. (2007)*. On the other hand, *Zhang et al. (in press)* used the RT-LAMP as a rapid and quantitative diagnostic assay for the detection of CMNV in *P. vannamei*. This assay is capable of detecting as little as 6.3 pg of total RNA from infected shrimps.

Assays based on lateral flow strips have also been deployed for diagnosis of nodavirus. *Lin et al. (2014)* combined RT-LAMP with a lateral flow dipstick (RT-LAMP-LFD) for the detection of *Mr*NV, targeting six distinct regions of *Mr*NV RNA2. The sensitivity of this RT-LAMP-LFD is 10x higher than that of the RT-LAMP. *Toubanaki, Margaroni & Karagouni (2015b)* also developed a lateral flow paper biosensor for the detection of NNV in European seabass. Instead of RT-LAMP, this lateral flow biosensor detects the viral RNA through RT-PCR using a 5′-biotin-tagged primer, a probe containing a poly-A tail, and gold nanoparticles conjugated to a poly-T oligonucleotide, with streptavidin forming the test line. This assay was reported to detect 270 pg of initial total RNA, which is less sensitive than the RT-LAMP based method (6.3 pg).

Apart from simply detecting the presence of nodavirus infection, it is also important to identify the genotype of the infecting virus, either for epidemiological study, or for specific strategy design to eliminate virus infection in an aquaculture farm. *Toubanaki, Margaroni & Karagouni (2015a)* developed a tetra-primer PCR which can amplify specifically RGNNV

or SJNNV cDNA, thereby generating short PCR products of different sizes which can distinguish between RGNNV and SJNNV infections in European seabass.

Apart from detecting the virus at RNA level, the presence of nodavirus can also be evaluated by immunological methods, such as Western blotting, indirect florescent antibody, enzyme-linked immunosorbent assay (ELISA) and immunodot blot tests (*Ghiasi et al., 2016*; *Hegde et al., 2003*; *Sri Widada et al., 2003*). *Mr*NV infection has been diagnosed with Western blotting, dot blot and ELISA using polyclonal antibodies against the recombinant *Mr*NV capsid protein raised in rabbit (*Farook et al., 2014a*). *Wang, Chang & Wen (2016)* used an immunodot blot assay to detect *Mr*NV with a polyclonal antibody raised against the recombinant viral capsid in a Wistar rat. In addition, *Wangman et al. (2012)* successfully produced monoclonal antibodies that bind specifically to *Mr*NV capsid protein. These antibodies can be used to detect *Mr*NV without cross-reaction with other common shrimp viruses. Although the immunological methods are less sensitive compared with the viral RNA-based detection methods, the former remains a viable alternative for many laboratories.

### *In vitro* model for nodavirus studies

Cell lines are important models in virology, toxicology and gene expression studies. In virology, cell lines have been widely used to determine the infectivity, pathogenicity and infectious mechanisms of nodavirus (*Abdul Majeed et al., 2013*; *Nishi et al., 2016*). Currently, *Channa striatus* kidney (CSK) (kidney of *Channa striatus*), GB (brain of *Epinephelus coioides*), GF-1 (fin of *Epinephelus coioides*), SSN-1 (fry of *Ophicephalus striatus*), E-11 (clone of SSN-1), SISK (kidney of *Lates calcarifer*), SISS (spleen of *Lates calcarifer*), SIGE (eye of *Epinephelus coioides*), ICF (fin of *Clarias batrachus*), IEE (eye of *Etroplus suratensis*), IEG (gill of *Etroplus suratensis*), IEK (kidney of *Etroplus suratensis*), and IGK (kidney of *Epinephelus coioides*) fish cell lines were proven susceptible to nodavirus infections and thus suitable for *in vitro* propagation and studies of the viral infectious mechanisms (*Abdul Majeed et al., 2013*; *Chi, Hu & Lo, 1999*; *Sarath Babu et al., 2013*; *Kai, Wu & Chi, 2014*; *Nishi et al., 2016*). Among these cell lines, SISK, SISS and SIGE were found to be more susceptible to nodavirus infections and are thus suitable models for nodavirus propagation, diagnostic reagent and vaccine productions (*Sarath Babu et al., 2013*).

### Immune response against nodavirus infection

Immunity plays an important role in the prevention and recovery of nodavirus infection in aquatic animals. Overall, activation of innate immunity (such as NK cells and antimicrobial peptides), cellular T cell type I interferon immunity, and humoral immunity (immunoglobulins antibodies) cooperatively prevent nodavirus infection (*Chen, Wang & Chen, 2014*; *Costa & Thompson, 2016*). Nodavirus mainly affects fishes at larval stage, which may be due to the lack of well-developed adaptive immune cells as present in adult fishes that restrict the viral replication, thus minimize the development of pathological and clinical signs (*Overgård et al., 2012*).

Pathogen-associated molecular patterns (PAMPs) on RNA viruses are first recognized by pattern-recognition receptors (PRRs). This process subsequently induces intracellular

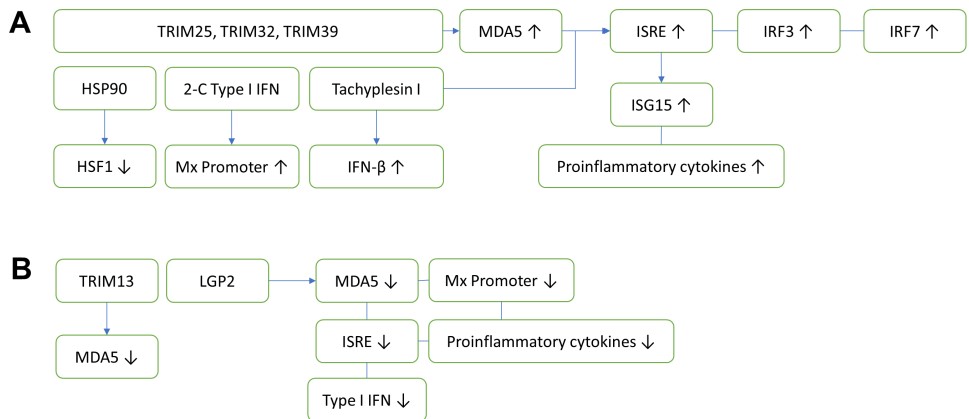

**Figure 1  Innate antiviral immunity of fish against nodavirus infection.** (A) Positive regulation which inhibits viral replication. TRIM25, TRIM32 and TRIM39 upregulate the expression of MDA5, which in turn induces ISRE, IRF3 and IRF7. The upregulation of ISRE also induces the expression of ISG15 and proinflammatory cytokines, cooperatively reducing the viral load. Other elements known to inhibit virus replication include HSP90, 2-C Type I IFN and Tachyplesin I, which downregulates HSF1, upregulates Mx promoter and IFN-β, respectively. (B) Negative regulation which promotes viral replication. TRIM13 and LGP2 downregulate MDA5, thereby reduce the ISRE. LGP2 also downregulates Mx promoter, proinflammatory cytokines and Type I IFN. TRIM, Tripartite motif-containing protein; MDA5, Melanoma differentiation-associated gene 5; IRF, Interferon regulatory factor; ISRE, Interferon-stimulated response element; HSF1, Heat shock transcription factor 1; HSP90, Heat shock protein 90; ISG, Interferon-stimulated gene.

signals to activate defensive mechanisms (*Chen et al., 2014*). Toll-like receptors (TLRs) and retinoic acid-inducible gene I (RIG-I)-like receptors (RLRs) are the two important classes of PRRs that sense the PAMPs of RNA viruses such as fish nodaviruses (*Chen et al., 2014*; *Costa & Thompson, 2016*). The RLR family consists of RIG-1, melanoma differentiation-associated gene 5 (MDA-5) and Laboratory of Genetics and Physiology 2 (LGP2). Activation of RLRs and TLRs subsequently promotes interferon type I antiviral immune response (*Costa & Thompson, 2016*). RLRs, MyD88-dependent TLRs (*Chen et al., 2015*) and TLR7 (*Takano et al., 2011*) were found upregulated and correlated with the production of type I interferon and pro-inflammation response. MDA5 is a member of RLRs that promotes transcription of interferon related immune factors, which include interferon regulatory factor 3 (IRF3), IRF7 (*Yu et al., 2017*), and interferon-stimulated response element (ISRE) such as interferon-stimulated gene 15 (ISG15) (*Huang et al., 2013*) and proinflammatory cytokines (*Huang et al., 2016b*). Unlike RIG-1 and MDA-5, the exact functions of LGP2 in different virus infections are controversial. LGP2 is generally reported as a positive regulator of RIG-1 and MDA-5 (*Chen et al., 2014*). However, a recent study showed a contrary function of LGP2, in which the overexpression of LGP2 suppressed the expression of MDA5, Mx promoter, ISRE, pro-inflammatory cytokines and type I interferon genes, thus resulting in a high viral load (*Yu et al., 2016*).

Tripartite motif-containing (TRIM) proteins are multi-domain proteins that exert important immune regulatory roles on TLR and RLRs mediated antiviral innate immunity (*Huang et al., 2016a*). The innate antiviral immunities of fish against nodavirus infection are summarized in Fig. 1. In fish, TRIM proteins play an important role in the recognition

and initiation of protection against nodavirus infection. To date, fish TRIM25, TRIM32 and TRIM39 exerted positive antiviral responses via activation of MDA5 expression (*Wang et al., 2016*; *Yang et al., 2016*; *Yu et al., 2017*). On the other hand, TRIM13 exerted negative regulation on antiviral immunity against nodavirus infection through downregulation of MDA5 and the downstream IFN signaling pathway (*Huang et al., 2016a*).

Type I (α/β) and type II (γ) interferons play important roles in the innate immune responses against nodavirus infection in fish. Post detection by PRRs, IFN type I will be secreted and picked up by IFN receptor of neighboring cells, activating its Janus kinases 1 (JAK1)/signal transducer and activators of transcription (STAT) pathway which leads to the transcription of IFN-stimulated genes (ISGs) (*Chen et al., 2014*) and pro-inflammatory cytokines, such as IL-1β, IL-6 and TNF-α (*Costa & Thompson, 2016*). Activation of ISGs via type I interferon subsequently promotes Mx promoter activity, which increases host resistance to nodavirus infection (*Chen et al., 2014*). Mx is one of the downstream antiviral effectors under type I interferon immunity (*Sadler & Williams, 2008*). The researchers demonstrated that the sevenband grouper *Epinephelus septemfasciatusu* pre-treated with non-lethal aquabirnavirus (ABV) developed a protection against the RGNNV nodavirus infection prior to the activation of type I interferon. This protection was attributed to the overexpression of the type I interferon downstream effector, the Mx gene in head kidney and brain of the fish (*Pakingking Jr et al., 2005*). Furthermore, inoculation of gilthead seabream *Sparus aurata* with lipopolysaccharides from *Vibrio alginolyticus* also stimulated the Mx gene expression in liver, which effectively reduced the load of nodavirus in the brain (*Bravo et al., 2013*).

Halibuts and groupers are known reservoirs of nodavirus but resistant to the virus and vertically transmit the disease (*Chaves-Pozo et al., 2012*; *Overgård et al., 2012*). Although the early proinflammatory type 1 interferon response helps to control the nodavirus infection in the juvenile Atlantic halibut *Hippoglossus hippoglossus*, the viral RNA was still detected in the brain, eye and head kidney of the fish even after 14 weeks post infection by the virus. This was accompanied by more drastic T cell mediated responses including upregulation of the T cell markers (CD4, CD8α and CD8β), ISG15, Mx and IFNγ genes expression (*Overgård et al., 2012*). However, elevation of proinflammatory cytokines without significant changes in CD4, CD8α and CD8β T cell markers was observed in the brain, eye and head kidney of Atlantic halibut at the early stage of nodavirus infection (*Overgård et al., 2012*). On the other hand, susceptible species such as European seabass were detected with delayed but stronger proinflammatory response, resulting in an irreparable brain damage (*Poisa-Beiro et al., 2008*; *Valero et al., 2015b*). In addition, a challenge study on sensitive seabass species with low titer of nodavirus induced early but short-term type I interferon response (*Scapigliati et al., 2010*). Moreover, *Valero et al. (2015a)* showed that nodavirus replicated more in the reservoir seabream's testis than the susceptible seabass, through modulation of reproductive system that favor the transmission and shedding of the virus in the reservoir species. This result indicates that proinflammatory type I interferon did not involve in stimulating T cell proliferation at the early stage of the viral infection, and thus T cell type I interferon response is not sufficient to clear the nodavirus, resulting in vertical virus transmission in the reservoir species. The researchers proposed that antimicrobial peptides

(AMPs) play an important role in vertical transmission of nodavirus in resistant fish species (*Valero et al., 2015b*). AMPs are a major component of the innate immune system in fish that activate antiviral effects upon nodavirus infection (*Xie, Wei & Qin, 2016*; *Valero et al., 2015b*). Grouper epinecidin-1 (CP643-1), complement factor 3 (c3), lysozyme (lyz), hepcidin (hamp), dicentracin (dic), piscidin (pis) or b-defensin (bdef) are AMPs found to be activated during nodavirus infection in both susceptible and resistant fish species (*Valero et al., 2015b*). CP643-1 was also reported to induce the Mx gene expression during nodavirus infection in fish (*Chia et al., 2010*). In addition, Tachyplesin I has been reported as an AMP found in resistant grouper strains which activates the antiviral activity through promotion of ISRE and IFN-β expression (*Xie, Wei & Qin, 2016*). Histones (H1 to H4) are another type of potential AMPs that may play some roles in the antiviral effects in fish. However, more studies are needed to investigate the function of histones in protecting fish against nodavirus infection (*Valero et al., 2016a*). Production of AMPs in resistant gilthead seabream and susceptible European seabass differs significantly, in which the AMPs were highly expressed in the brain but low in the gonad of gilthead seabream, whereas in European seabass it was highly expressed in the gonad but low in brain. These results indicate that vertical transmission of nodavirus by the resistant gilthead seabream could be attributed to the poor AMP response in the gonad. The European seabass containing a high level of AMPs in the gonad did not survive nodavirus infection as the AMP expression level in the brain was low (*Valero et al., 2015b*).

There are other immune factors contribute to the protection of fish against nodavirus infection. *Esteban et al. (2013)* reported that nodavirus strain 411/96 (RGNNV) induced early (day 1 post-challenged) expression of the peroxiredoxin natural killer enhancing factor A (NKEF-A), which involved in inflammation and innate immunity in both the brain and head kidney of gilthead seabream, but not in European seabass. This result shows that an early expression of NKEF-A which activates immune cells including the NK cells and macrophages is an important anti-nodavirus mechanism in resistant species. On the other hand, the involvement of CD83 gene in the immune response of fish during nodavirus infection was also evaluated. Downregulation of CD83-like molecule expression was observed in the head kidney of European seabass post-infected with nodavirus. Although CD83 was known as a marker for matured human dendritic cells, active thymic T cells and even B cells, the exact function of CD83 in fish lymphocytes is still unknown (*Buonocore et al., 2012*). Thus, more studies have to be performed to investigate the involvement of CD83 expression in immunity against nodavirus infection.

Nodavirus is a simple RNA virus with only three genes. However, it has developed some virus-host interaction properties, which include hijacking the host system and escaping host defense mechanism (*Chen et al., 2014*). Overexpression of heat shock transcription factor 1 (HSF1) promoted the replication of nodavirus at the initial stage of the viral infection, which could be due to an increase of fish body temperature as the expression of Mx protein was suppressed at high temperature condition (*Wang, Chen & Chen, 2016*). Suppression of HSF1 by the heat shock protein 90 (HSP90) thereby reduced the replication of nodavirus during the initial stage of the viral infection (*Chen et al., 2010*; *Wang, Chen & Chen, 2016*). Moreover, fish is more susceptible to nodavirus infection in the present of

immunosuppressive agents. *Lawrence, Reid & Whalen (2015)* reported that an organotin compound, tributyltin, which is commonly used as antifouling paints for ships and fishing nets caused immune suppression in fishes. Exposure of Japanese medaka *Oryzias latipes* larvae to tributyltin increased their susceptibility towards SGWak97 nodavirus (RGNNV) infection, which resulted in a higher mortality in a dosage dependent manner. This phenomenon could be attributed to the immunosuppression caused by tributyltin on fish NK cell activity (*Kitamura et al., in press*).

Besides innate and cell mediated immunity, humoral immunity also plays an important role in protecting fish against nodavirus infection (*Chen et al., 2014*), especially the two immunoglobulins, IgM and IgT. Infection of susceptible seabass with low titer of nodavirus did not significantly alter the expression of IgM and IgT in the gills and spleen (*Buonocore et al., 2017*) but only induced a marginal increase of serum IgM (*Scapigliati et al., 2010*). On the other hand, activation of IgM$^+$ and IgT$^+$ B cells in the brain and overexpression of soluble IgT$^+$ by B cells in the head kidney by early inflammatory response in the central nervous system reduced nodaviral replication in the resistant aquaculture-relevant fish species (*Lopez-Munoz et al., 2012*; *Piazzon et al., 2016*). Thus, activation of IgM and IgT expression by vaccination can protect the fish from nodavirus infection (*Costa & Thompson, 2016*).

Unlike vertebrates including fish, the understanding of prawn immunity against nodavirus infection is even limited. Based on current findings, prawns generally fight infections through non-specific innate immune responses including prophenoloxidase-activating system (*Ourth & Renis, 1993*; *Popham et al., 2004*) and over-accumulation of superoxide anion (*Ravi et al., 2010*), which were known to inactivate DNA and RNA viruses. However, the basic understanding on the prawn immunity against nodavirus infection has ignited a spark of interest among researchers to produce vaccines against the prawn nodavirus. Instead of the whole virus, *Farook et al. (2014b)* introduced a recombinant *Mr*NV capsid protein (r-MCP) produced in *E. coli* into *M. rosenbergii* as a potential vaccine against the WTD. This r-MCP increased the level of prophenoloxidase, superoxide anion, and other anti-viral compounds such as crustin, peroxinectin, anti-lipopolysaccharides and heat shock proteins (HSP21, HSP70, HSP90), which protected the *M. rosenbergii* post-larvae from *Mr*NV challenge up to 76%.

## Advances in nodavirus vaccine development

Vaccination has been proposed as a solution to control and prevent nodavirus outbreaks in aquaculture industry (*Pakingking Jr et al., 2010*). Table 1 summarizes the studies on nodavirus vaccines. Among different types of vaccines, virus-like particles (VLPs) show the highest potential to induce a long lasting and protective humoral immunity (*Liu et al., 2006b*). Intramuscular administration of recombinant DGNNV VLPs produced in *E. coli* induced a high antibody titer which is capable to neutralize the virus *in vitro*. Even without an adjuvant, neutralizing antibodies induced by the DGNNV VLPs lasted over five months, further justifying the potential application of nodavirus VLPs as a vaccine. A recombinant betanodavirus capsid protein r-FNCP42 was generated by *Vimal et al. (2014)* based on the gene sequence of a fish nodavirus isolated from Asian seabass (*L. calcarifer*) larvae.

**Table 1** Vaccines, route of administration and their protectivity.

| Type of vaccine | Route of vaccination | Protectivity | Remarks | References |
|---|---|---|---|---|
| Recombinant betanodavirus of RNA2 capsid protein r-FNCP42 | IM | 75% higher survival rate of juveniles of Asian seabass challenged with $1 \times 10^{6.5}$ TCID$_{50}$ of nodavirus/fish | As the genome sequence analysis of r-FNCP42 has more than 98–99% of similarity with other fish nodavirus including red spotted grouper nervous necrosis virus, Dicentrarchus labrax encephalitis virus, Asian seabass nervous necrosis virus, and Epinephelus tauvina nervous necrosis virus (ETNV), thus cross protectivity of r-FNCP42 against other strains of nodavirus shall be tested. | *Vimal et al. (2014)*;*Vimal et al. (2016)* |
| Recombinant r-FNCP42-DNA | IM | 77% higher survival rate of juveniles of Asian seabass challenged with $1 \times 10^{6.5}$ TCID$_{50}$ of nodavirus/fish | Capsid protein was highly expressed in the heart, muscle and liver of the vaccinated fish. | *Vimal et al. (2016)* |
| Recombinant capsid protein MGNNV virus like particles (VLPs) | IM | $\sim$70% higher survival rate of juvenile European seabass (Dicentrarchus labrax) challenged with $10^5$ TCID$_{50}$/fish | MGNNV induced humoral immunity against nodavirus. | *Thiery et al. (2006)* |
| DNA vaccine pVHSV-G encoding glycoprotein of viral hemorrhagic septicaemia virus | IM | $\sim$54% higher survival rate of juvenile turbot (Scophthalmus maximus) challenged with $10^{6.3}$ TCID$_{50}$ | DNA vaccine induced inflammatory response that cross protect nodavirus infection. | *Sommerset et al. (2003)* |
| Synthetic peptides (N-terminal regions) of nodavirus DIEV RNA2 protein | IM | $\sim$27% higher survival rate of seabass challenged with $10^9$ FCU/fish | Peptides induced humoral immunity. | *Coeurdacier, Laporte & Pepin (2003)* |
| Heat inactivated S1 and Sb2 nodavirus | IM | $\sim$33% and 26% higher survival rate of seabass challenged with $9 \times 10^9$ FCU/fish, respectively | Induced humoral immunity. | *Coeurdacier, Laporte & Pepin (2003)* |
| Virus-like particles (VLPs) of grouper nervous necrosis virus | IM | – | Induced humoral immunity. No challenge test was performed. | *Liu et al. (2006b)* |
| Recombinant RGNNV-CP | IM | $\sim$60% higher survival rate of humpback grouper challenged with $10^{5.5}$ TCID$_{50}$/fish, respectively | Induced humoral immunity. | *Yuasa et al. (2002)* |
| Recombinant ETNNV-CP (Epinephelus tauvina nervous necrosis virus-capsid protein) | IM | – | Induced stronger humoral immunity than formalin inactivated nodavirus. No challenge test was performed. | *Hegde, Lam & Sin (2005)* |
| Formalin inactivated nodavirus | IP | 60% higher survival rate of brown-marbled grouper challenged with $10^{6.5}$ TCID$_{50}$/fish of OSGBF1E | Induction of humoral immunity. | *Pakingking Jr et al. (2009)* |

**Table 1** (*continued*)

| Type of vaccine | Route of vaccination | Protectivity | Remarks | References |
|---|---|---|---|---|
| Recombinant capsid protein, recAHNV-C | IP | 29% higher survival rate of juvenile turbot (Scophthalmus maximus) challenged with $10^6$ TCID$_{50}$/ml AHNV | Fishes vaccinated with plasmid DNA expressing the recombinant capsid protein were not protected as the plasmid DNA only induced cellular but not humoral immunity. | *Sommerset et al. (2005)* |
| Formalin inactivated SGWak97 | IP | Not reported | Inactivated SGWak97 induced humoral immunity. | *Pakingking Jr et al. (2009)* |
| Recombinant rT2 SJNNV-CP (Scophthalmus maximus nervous necrosis virus-capsid protein) | IP | ∼36% higher survival rate of humpback grouper challenged with $6.3 \times 10^7$ TCID$_{50}$/fish, respectively | Induced humoral immunity. | *Húsgağ et al. (2001)* |
| Chitosan-encapsulated DNA vaccine (CP-pNNV) | Oral | 55% higher survival rate of juvenile European seabass (Dicentrarchus labrax) challenged with $10^6$ TCID$_{50}$/fish | CP-pNNV failed to induce humoral immunity but activated interferon pathway and cell-mediacted cytotoxicity. | *Valero et al. (2016b)* |
| Chitosan conjugated DNA vaccine pcDNA-XSVAS | Oral | Approximately 50% higher survival rate of prawn challenged with crude extract of prawn with WTD. | XSV with nodavirus caused white tail disease (WTD) in prawn. The challenge experiment shall consider using isolated virus instead of crude one. | *Ramya et al. (2014)* |
| Recombinant yeast expressing RGNNV-CP (red-spotted grouper necrosis virus capsid protein) | Oral | – | Induced humoral immunity in mice. No challenge test was performed. | *Kim et al. (2013)* |
| Artemia-encapsulated recombinant pET24a-NNV VP *E. coli* expressing nodavirus capsid protein | Oral | ∼34% higher survival rate of grouper larvae challenged with $10^5$ TCID$_{50}$/fish, respectively | Induced humoral immunity. | *Lin et al. (2007)* |
| Inactivated bacteria encapsulated dsRNA of *Mr*NV and XSV | Oral | *Mr*NV challenge 24 h and 72 h post-feeding showed relative percent survival of 80% and 75%, respectively | Protection through RNA interference with capsid and B2 genes of *Mr*NV, and capsid gene of XSV. | *Naveen Kumar, Karunasagar & Karunasagar (2013)* |
| Solid lipid nanoparticles encapsulated binary ethylenimine inactivated nodavirus | Bath and Oral | 45% higher survival rate of grouper larvae challenged with $1 \times 10^6$ TCID$_{50}$/ml HGNNV | Simple vaccination procedure that fit for larvae. Both routes of vaccinations induced pro-inflammatory cytokines expression, type I IFN response, humoral immunity and cellular immunity. | *Kai & Chi (2008), Kai, Wu & Chi (2014)* |
| Recombinant *Mr*NV capsid protein | Bath | Immersion for 24 h followed by *Mr*NV challenge showed 76.03% survival in 15 days post-challenge | Protection is believed to be through upregulation of prophenoloxidase, superoxide anion and SOD activity. | *Farook et al. (2014b)* |

**Notes.**

IM, intramuscular injection; IP, intraperitoneal injection; Oral, oral feeding; Bath, immersion.

Intramuscular injection of 50 μg r-FNCP42/fish resulted in 75% survival of juveniles of Asian seabass challenged with $1 \times 10^{6.5}$ TCID$_{50}$ of nodavirus. As the genome sequence of r-FNCP42 shares more than 98–99% similarity with other fish nodaviruses including red spotted grouper nervous necrosis virus, *Dicentrarchus labrax* encephalitis virus, Asian seabass nervous necrosis virus, and *Epinephelus tauvina* nervous necrosis virus (ETNV), thus cross protectivity of r-FNCP42 against these nodaviruses should be tested. *Naveen Kumar, Karunasagar & Karunasagar (2013)* immunized *M. rosenbergii* through oral administration of inactivated bacteria encapsulated dsRNA of *Mr*NV and XSV, where a post-feeding virus challenge showed promising results. The *Mr*NV challenge at 24 h and 72 h post-feeding showed relative high percentage of survival at 80% and 75%, respectively, indicating a regulation via RNA interference. *Ramya et al. (2014)* used chitosan conjugated DNA vaccine, where XSV antisense (XSVAS) nucleotide sequence was cloned into the pcDNA plasmid vector. The presence of plasmid pcDNA-XSVAS was confirmed after 30 days of administration through oral feeding, where it provided approximately 50% protection to prawns challenged with crude extract of WTD-prawns. In addition, introduction of recombinant *Mr*NV capsid protein through 24 h immersion followed by *Mr*NV challenge boosted the relative percent survival of prawns by 76.03% (*Farook et al., 2014b*).

## Virus-like particles

After decades since nodaviruses were first discovered, studies on their VLPs continue. DGNNV VLPs produced in *E. coli* were crystallized and studied with x-ray diffraction, revealing a $T = 3$ icosahedral structure approximately 38 nm in diameter, closely resembling the native virion (*Luo et al., 2014*). Recombinant FHV capsid protein produced in *E. coli* was also used in an *in vitro* assembly study (*Bajaj & Banerjee, 2016*). The capsid protein possesses additional N-terminal tag which hinders the assembly of the capsid protein into VLPs. Cleavage of this N-terminal region *in vitro* in the presence of $Ca^{2+}$ ion allows the capsid protein to assemble into VLPs of different sizes. Despite the heterogenicity, these VLPs were capable of membrane disruption, a property required by the nodavirus to penetrate its host cells (*Bajaj & Banerjee, 2016*).

In addition, FHV VLPs have also been used to display foreign epitopes, such as that of hepatitis C virus (HCV) (*Peng, Dai & Chen, 2005*). *Chen et al. (2006)* used FHV VLPs to display the epitopes of HCV core protein and hepatitis B virus (HBV) surface antigen, where the displayed epitopes were shown to be immunogenic in guinea pigs. In another study, *Manayani et al. (2007)* fused the protective antigen-binding von Willebrand A domain of ANTXR2 cellular receptor to FHV VLPs. The researchers demonstrated that the fusion protein inhibited lethal anthrax toxin, and at the same time induced toxin-neutralizing antibody which protected rats from anthrax lethal toxin challenge. All of these studies demonstrated the potential of insect nodavirus VLPs as a foreign epitope presenting agent. In recent years, however, studies on nodavirus VLPs focused more on the prawn nodavirus, particularly *Mr*NV.

The first prawn nodavirus VLPs were produced by *Goh et al. (2011)* via recombinant DNA technology. The recombinant *Mr*NV capsid protein expressed in *E. coli* self assembles

into VLPs of approximately 30 nm in diameter (*Goh et al., 2011*). Recently, they reported that 20–29 amino acids (a.a.) at the N-terminal region of *Mr*NV capsid protein are responsible for RNA binding during the VLPs assembly through ionic interaction, where mutation of positively charged a.a. at this region to alanine abolished the RNA binding of the *Mr*NV capsid protein (*Goh et al., 2014*). Despite the role of RNA binding, the N-terminal region (1–29 a.a.) is not required for the assembly of the VLPs, as demonstrated by *Goh et al. (2014)*. *Jariyapong et al. (2014)* demonstrated the ability of the *Mr*NV VLPs to encapsidate plasmid DNA in 0.035–0.042 mol ratio (DNA/ protein) through particle disassembly and reassembly, with the use of EGTA (ethylene glycol-bis(2-aminoethylether)-N,N,N′,N′-tetraacetic acid) and $Ca^{2+}$ ion, thereby opening a path for the *Mr*NV VLPs to be used for the delivery of nucleic acid based therapeutic agents, such as DNA vaccine or siRNA.

VLPs have been widely used for displaying foreign epitopes, for instance the VLPs of human papilloma virus (*Matic et al., 2011*), HBV (*Ibañez et al., 2013*; *Murray & Shiau, 1999*; *Yap et al., 2012*), as well as bacteriophages (*Hashemi et al., 2012*; *Kok et al., 2002*; *Tan et al., 2005*; *Wan et al., 2001*). VLPs are known to enhance the immunogenicity of small epitopes displayed on the particles (*Murata et al., 2003*; *Quan et al., 2008*). We have displayed the immunodominant region of HBV on the surface of the *Mr*NV VLPs through fusion at the C-terminal end of *Mr*NV capsid protein and confirmed the fusion protein with immunogold TEM (*Yong et al., 2015a*). When introduced into BALB/c mice, this recombinant VLPs induced the production of anti-HBV antibodies, as well as the cellular immune responses including natural killer cells, cytotoxic T lymphocytes (CTL) and IFNγ. In addition, we have fused and displayed multiple copies of influenza A virus matrix 2 ectodomain (M2e) on the surface of the *Mr*NV VLPs (*Yong et al., 2015b*). The displayed M2e epitopes were highly antigenic and immunogenic, where they correlated well with the copy number of M2e displayed on the surface of the VLPs. Most recently, *Somrit et al. (2017)* showed that the C-terminal region of *Mr*NV capsid protein is exposed on the surface of the VLP, constituting the core of the viral capsid protrusion, through a homology-based modeling based on cucumber necrosis virus. When the C-terminal region was removed with chymotrypsin digestion, the internalization capability of the truncated VLPs into Sf9 cells reduced significantly, suggesting the importance of the C-terminal region in the viral infection.

In an attempt to discover the possible mechanism of *Mr*NV infection pathway, we used the *Mr*NV VLPs labeled with fluorescein as a model to study the *Mr*NV entry and localization in Sf9 cells (*Hanapi et al., 2017*). Through the use of endosomal inhibitors coupled with laser confocal microscopy and live cell imaging, we demonstrated that the internalization of *Mr*NV VLPs was facilitated by clathrin- and caveolae-mediated endocytosis. We have also identified a potential nuclear localization signal (NLS), which could aid in the localization of *Mr*NV capsid protein to the nucleus based on the importin-α pathway (*Hanapi et al., 2017*).

Apart from the *Mr*NV VLPs produced in *E. coli*, we have also produced the *Mr*NV capsid protein in Sf9 cells through baculovirus expression system (*Kueh et al., 2017*). This eukaryotic produced *Mr*NV capsid protein self-assembles into VLPs significantly larger than their prokaryotic counterparts. The Sf9 produced *Mr*NV VLPs are structurally

more homogenous as observed by TEM, representing a better candidate to be used in structural study. Subsequently, we used this *Mr*NV VLPs produced in Sf9 for 3D structure reconstruction using the images obtained from cryogenic electron microscopy (*Ho et al., 2017*). The 3D structure of *Mr*NV capsid at 7 Angstroms resolution reveals a $T = 3$ icosahedral structure distinctive to other insect and fish nodavirus capsids, characterized by large dimeric blade-like spikes exposed on the surface of the VLPs. This finding supports the assertion that prawn nodavirus should be classified into a new genus.

## CONCLUSIONS

Prawn nodaviruses are relatively new compared to the typical alpha- and beta-nodaviruses. Despite their genomic and structural differences with the two established genera, prawn nodaviruses have yet been classified into a new genus. There are likely more prawn nodaviruses unknown to men, such as the recently discovered CMNV and FdNV. Isolation and characterization of these new prawn nodaviruses could contribute in creating a new genus of *Nodaviridae*, which is the gamma-nodavirus. In addition, there is not yet an effective mid- or long-term vaccine for shrimps and prawns against nodavirus infections. Due to the lack of adaptive immune response in crustacean, antigens that can induce protection against the infections have to be administrated from time to time, especially during the larval stage. Therefore, nodavirus vaccines based on recombinant proteins incorporated into feeds would be more relevant. Better yet, self-replicating DNA expression vector-based vaccines would be more cost-effective to be utilized in shrimp and prawn aquaculture industries.

### Funding
This study was supported by the Ministry of Science, Technology and Innovation, Malaysia (grant no: 02-01-04-SF2115) and UPM (grant no: GP-IPS/2016/9511000). The funders had no role in study design, data collection and analysis, decision to publish, or preparation of the manuscript.

### Grant Disclosures
The following grant information was disclosed by the authors:
Ministry of Science, Technology and Innovation, Malaysia: 02-01-04-SF2115.
UPM: GP-IPS/2016/9511000.

### Competing Interests
The authors declare there are no competing interests.

### Author Contributions
- Chean Yeah Yong and Swee Keong Yeap planned the contents of the manuscript, performed data searches, analyzed the data, wrote the paper, and prepared the figure and table.

- Abdul Rahman Omar analyzed the data, contributed analysis tools, wrote the paper, and reviewed drafts of the paper.
- Wen Siang Tan planned the contents of the manuscript, collect the data, analyzed the data, contributed analysis tools, and wrote the paper.

### Data Availability

The research in this article did not generate any data or code (literature review).

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
