# Peer review of "Advances in the study of nodavirus"

_PeerJ, doi:10.7717/peerj.3841_

## Round 0.1 · original submission · Major Revisions

Some points have to be revised and improved. Please follow the guidelines provided by the reviewers, both of whom make quite similar points.

·

Basic reporting

In overall, the manuscript was written with clear and unambiguous English. There are some points that need to be addressed. The references were sufficient.

Experimental design

This manuscript is a review article.

Validity of the findings

This manuscript is a review article.

Additional comments

Advances in the study of nodaviruses
This review manuscript described nodaviruses in various aspects including transmission, classification, immune response and diagnosis using molecular and immunological techniques. Overall, the manuscript thoroughly explain in detail; however, there are some points that should be added for completeness. The whole M.S. should be edited by native English speaker.
1. Line 86: The RGNNV has been shown to infect tilapia (Keawcharoen,J, et al., J. Fish Dis. 38, 49-54 (2015). This information should be added.
2. Line 293: The monoclonal antibody specific to MrNV was reported (Wangman et al., Dis. Aquat. Org. 2012, 98(2):121-31). The information based on this report should be mentioned.
3. Line 143: What is a well-established cell line? The detail is needed.

Minor points
- The “Mr” in “MrNV” should be italicized throughout the manuscript.
- Lines 27, 207: use “Histopathology..”
- Line 75-76: use “…….infected fishes. Neither alpha-nor beta-nodavirus………(NaveenKumar et al., 2013). Another type of nodavirus includes…….
- Line 78: use “…PvNNV should be categorized…”
- Line 79: use “…compare with that of both….”
- Line 165: use comprise of
- Line 167: use “Both of which ….”
- Line 213: use “lymphocytes”
- Line 225: use “…with that of members…”
- Line 246: italicize “in situ” and use “..nested RT-PCR…”
- Line 274: use “..higher than that of …”
- Line 279: use “…lower than that of …”
- Line 422: change “vertebrate” to “fishes”

Reviewer 2 ·

Basic reporting

This review is well-written.
Literature should be implemented.

Experimental design

Not applicable.

Validity of the findings

Not applicable.

Additional comments

The manuscript is a review about Nodavirus. Paper is well written and clear.
Respect to the first statements about recent reviews in the field there are some good reviews not included covering distinct aspects:
Doan QK, Vandeputte M, Chatain B, Morin T, Allal F. Viral encephalopathy and retinopathy in aquaculture: a review. J Fish Dis. 2017 May;40(5):717-742.
Costa JZ, Thompson KD. Understanding the interaction between Betanodavirus and its host for the development of prophylactic measures for viral encephalopathy and retinopathy. Fish Shellfish Immunol. 2016 Jun;53:35-49.
Reshi ML, Su YC, Hong JR. RNA Viruses: ROS-Mediated Cell Death. Int J Cell Biol. 2014;2014:467452.
Hong JR. Betanodavirus: Mitochondrial disruption and necrotic cell death. World J Virol. 2013 Feb 12;2(1):1-5.
Chen YM, Wang TY, Chen TY. Immunity to betanodavirus infections of marine fish. Dev Comp Immunol. 2014 Apr;43(2):174-83.
Chaivisuthangkura P, Longyant S, Sithigorngul P. Immunological-based assays for specific detection of shrimp viruses. World J Virol. 2014 Feb 12;3(1):1-10.
But also other reviews related to viral diseases, immunity, etc. in aquaculture species during the last decade. Thus, the starting point of this review is not adequate.
The topic is certainly interesting and merits further research since nodaviruses are actually producing probably the largest economic loss in the productive sector.
Regarding insect nodaviruses, there is only scarce explanation and this could be either deleted from the paper or reinforced with further information (detection, immunity, vaccines…).
Comments:
- Line 100. The number of infected fish species is much larger. Please, revise it.
- Line 104. Starting from this point include the latin name of all animal species in the text.
- Line 143. The cell line name is lacking.
- Line 320. This reference in not appropriate for this.
- Line 340. Classification of Mx as an AMP is not adequate and consistent. This belongs to the type I IFN response rather to the AMP repertoire.
- Section related to the immunity, both fish and prawns, needs to be improved and better organized. It is not clear whether the cited studies area about gene expression or functional and this is an important issue to be denoted. References in this section need to be further revised and better selected. For example, important references should be included:
doi: 10.1016/j.fsi.2009.11.008.
doi: 10.1016/j.dci.2012.02.007
doi: 10.1016/j.fsi.2011.03.018

---

## Round 0.2 · accepted · Accept

After reading your revised paper and see the reviewers it is my pleasure to accept your manuscript in its present form. Congratulations!

·

Basic reporting

The English is clear and unambiguous.

Experimental design

The review included many updated data.

Validity of the findings

The conclusion was drawn as shown in Fig. 1.

Additional comments

All of the questions were properly responded. I have no further comments.

Reviewer 2 ·

Basic reporting

The paper is OK.

Experimental design

NA

Validity of the findings

Ok

Additional comments

The manuscript has been improved accordiingly.